# Machine Vision-Assisted Design of End Effector Pose in Robotic Mixed Depalletizing of Heterogeneous Cargo

**DOI:** 10.3390/s25041137

**Published:** 2025-02-13

**Authors:** Sebastián Valero, Juan Camilo Martinez, Ana María Montes, Cesar Marín, Rubén Bolaños, David Álvarez

**Affiliations:** 1Department of Industrial Engineering, Universidad de los Andes, Bogotá 111711, Colombia; js.valero10@uniandes.edu.co (S.V.); jc.martinez10@uniandes.edu.co (J.C.M.); a.montesf@uniandes.edu.co (A.M.M.); 2Integra S.A., Pereira 660003, Colombia; cmarin@integra.com (C.M.); rbolanos@integra.com (R.B.)

**Keywords:** segmentation, hand–eye calibration, robotic depalletizing, heterogeneous cargo handling, deep learning in robotics, sparse stereo vision, 3D object detection, non-orthogonal pallet configurations, pick-and-place robotics

## Abstract

Automated depalletizing systems aim to offer continuous and efficient operation in warehouse logistics, reducing cycle times and contributing to worker safety. However, most commercially available depalletizing solutions are designed primarily for highly homogeneous cargo arranged in orthogonal configurations. This paper presents a real-time approach for depalletizing heterogeneous pallets with boxes of varying sizes and arbitrary orientations, including configurations where the topmost surfaces of boxes are not necessarily parallel to each other. To accomplish this, we propose an algorithm that leverages deep learning-based machine vision to determine the size, position, and orientation of boxes relative to the horizontal plane of a robot arm from sparse depth data. Using this information, we implement a path planning method that generates collision-free trajectories to enable precise box grasping and placement onto a production line. Validation through both simulated and real-world experiments demonstrates the feasibility and accuracy of this approach in complex industrial settings, highlighting potential improvements in the efficiency and adaptability of automated depalletizing systems.

## 1. Introduction

Automated depalletizing solutions consist of systematically removing goods or materials from a pallet to place them on a production line [1]. They can operate continuously, significantly reducing cycle times compared to manual methods and enhancing production rates. By performing repetitive tasks with high accuracy and consistency, these systems minimize human error and can handle hazardous or heavy materials, reducing the risk of workplace injuries. Furthermore, these systems can be reprogrammed to adapt to changes in production lines or to handle new products without the need for extensive structural reconfiguration [2].

Most existing depalletizing systems, such as the one demonstrated by Etde [3], perform well with homogeneous and consistently orthogonal pallet configurations, but exhibit increased error rates when these conditions are not fully met. This paper explores the design and implementation of an automated depalletizing system tailored for heterogeneous pallets, starting by presenting the general depalletizing scheme, which outlines the system configuration and operational workflow. Subsequently, we present a deep learning-based computer vision algorithm designed to locate boxes in three-dimensional space, accounting for their position and orientation, using sparse depth information obtained from passive stereo cameras. This approach enables scene understanding of pallets composed of tilted boxes. Following this, the process for calculating the end effector orientation is detailed, ensuring alignment with the tilted surfaces of the boxes for precise handling. Finally, the paper examines the motion planning of the robotic manipulator, emphasizing the development of safe and efficient trajectories to perform the depalletizing process.

The paper is structured in five sections, starting with a literature review in Section 2, where an overview of the depalletizing process is given, discussing the main system requirements, relevant research in the field as well as the identified knowledge gaps in pallet and package recognition. Section 3 details the proposed methodology and materials used in the system, including the definition of the work area of the robotic manipulator, the method used to identify box parameters (locations, orientations and sizes), an algorithm for estimating the end effector pose, and the proposed approach for path planning. Section 4 presents the results, including a description of the experimental testing of the prototype in a laboratory environment and a detailed analysis of these results. Finally, Section 5 draws the relevant conclusions, summarizes the impact of the proposed depalletizing solution, and proposes knowledge gaps that may be addressed in future works.

## 2. Literature Review

The literature review is organized into four subsections. The first subsection describes the automatic depalletizing process and examines its industrial applications. The second subsection focuses on computer vision, specifically areas focused on object detection and pose estimation in conditions typical of depalletizing scenarios. The third subsection discusses mechanical designs that support depalletizing solutions, such as gripper designs and elevated conveyor belts. Finally, we identify knowledge gaps relevant to this review, highlighting areas that justify our contributions.

### 2.1. Automated Depalletizing

Generally speaking, the automated depalletizing process is composed of a system that includes an autonomous mobile robot, a robotic arm, a lifting mechanism, and a visual sensor, as indicated by the authors in [4,5]. The literature suggests depalletizing systems are designed mainly to extract items from homogeneous pallets. Such is the case of the algorithm developed by [6], which was designed to stack packages in a staggered arrangement to ensure structural stability. Chiaravalli in [7] proposed a solution that employs a multisensor vision system and a force-controlled collaborative robot to detect the boxes on the pallet. More than one robot can be employed in the depalletizing process, as proposed by Mingyang [8], who showed an ABB industrial robot and a KUKA mobile manipulator, expanding the depalletizing area. Additionally, Aleotti [9] ensured the accurate execution of tasks by implementing force measurement and robot admittance control.

Some applications that highlight the importance of the depalletizing process can be found in the healthcare sector, as demonstrated by Opaspilai [10] using a robotic system to handle products before dispensing medication through a SCARA robot, or in the food sector, where Arpenti [11] presents an approach for the detection, recognition, and position estimation of boxes of various types in a robotic cell. Caccavale in [12] proposes a robotic depalletizing system specifically designed to integrate seamlessly into supermarket logistics processes and Kalapyshina presents an innovative robotic system specifically designed to depalletize glass containers in an automated food packaging environment, known as DP3 [13].

### 2.2. Depalletizing-Focused Machine Vision

Conventional computer and machine vision techniques face difficulties in quickly discerning packing patterns in heterogeneous stacking configurations, especially in scenarios with additional complications related to light, whether light is scarce, or there is overexposure due to an overly bright source of light, or glare artifacts due to direct lighting on objects with high reflectivity. Some of these complications have been addressed by Holz in [14], which presents a system designed for depalletizing that includes a comprehensive process for accurate object detection and localization through a multi-resolution surface model approach; by Kim [15], who proposes a method based on deep learning and image processing for efficient detection of cluttered and diverse packet surfaces, employing RGBD technology; and by Costa [16], who investigates the integration of artificial intelligence in industrial robots for automated depalletizing. To address scarce or unstable ambient lighting, Li [17] proposes a two-step method to detect relevant objects. First, the region-growing method is used to extract the whole target region from the original image. Then, a region proposal is achieved by the Probabilistic Hough Transform (PPHT). However, morphological operations required to implement the method proposed by Li are highly sensitive to parameter tuning, and offer poor adaptability.

### 2.3. Mechanical Grasping in Pallets

Precise end effector positioning on industrial robots is essential to ensure stability and accuracy during the handling of boxes on tilted pallets. In this context, large positioning errors can arise due to disturbances that are difficult to compensate for using conventional control methods [18]. In non-orthogonal environments, any misalignment in the position of the end effectors can lead to incorrect stacking, affecting cargo stability and increasing the risk of damage to the packages. Furthermore, precise positioning contributes to operational safety by reducing the likelihood of operator injury due to falling boxes. This technical aspect is therefore essential to improve efficiency and safety in automated handling operations in the industry.

Regarding the design of the grip for robotic handling of boxes, several strategies ensure robust and safe gripping of work objects, adapting to the geometry and properties of the object, as shown in [19,20]. Among the most common techniques, the lateral or pincer grip applies pressure on opposite sides of the box and is suitable for regular-sized objects; the suction grip, using suction cups, allows handling boxes with smooth surfaces [21]; and the top or clamp grip holds the box from the top, providing stability for medium or large objects. Also notable are the corner grip, which offers additional contact points for small boxes, and the base grip, ideal for lifting heavy boxes. In addition, multi-finger grippers provide adaptability by adjusting to different orientations and shapes [22], while the combined suction and pincer grip provide additional stability on irregular-sized objects. The choice of grip type depends on factors such as box size, weight, material, and weight distribution, as well as spatial restrictions [23] and even sensor accuracy [24].

To address low grasping accuracy and high failure rates, Guo et al. [25] designed a 3D vision-based robotic palletizing system using the Shape-NCC algorithm. Similarly, Fontanelli in [26] presented a flexible and adaptable gripper designed specifically for robotic depalletizing, thus facilitating the handling of complex-shaped boxes. For its part, Tanaka in [27] proposed a vacuum suction end effector designed specifically for depalletizing robots in distribution centers, offering increased load capacity. Similarly, the difficulties of optimizing the speed of depalletizing compared to the annual process have been observed, as presented by Eto in [28].

To the best of our knowledge, no other work addresses the depalletizing of heterogeneous pallets with tilted boxes in non-orthogonal arrangements, as detailed in [24]. Furthermore, the literature is limited concerning the incorporation of workspace filters to optimize inverse kinematics calculations, also highlighted in [16]. Another identified area of improvement in depalletizing processes is the consideration of only the width and length of the packages, obtained by simple observation of the topmost surfaces, discarding height information. This limitation, pointed out in [28], is addressed in this study.

## 3. Materials and Methods

### 3.1. Proposed Depalletizing Scheme

In the proposed scheme, a Universal Robots UR10 cobot is positioned on a platform near a conveyor belt and a movable pallet base. The UR10 is a serial manipulator, a type of robot that can reach various positions and orientations within a working area approximately defined by a spherical volume, as illustrated in Figure 1b.

A depth camera composed of two lateral infrared sensors and a central RGB sensor, located on the end effector of the manipulator, is positioned to capture the entire pallet to obtain its physical dimensions and location within the working space of the robotic manipulator. To estimate these values, it is necessary to move the robotic manipulator to the pallet identification pose. This pose will only be used once per depalletizing cycle. This same pose is used to determine the maximum height of the boxes per stratum of the pallet. Once the height of the pallet is known, the end effector of the robotic manipulator is positioned on the pallet, allowing a top view of the boxes. This pose will be called a box acquisition pose. Each time the end effector is positioned in the acquisition pose, the sensor scans for the pose of a selected box, and this information is processed to start the path-planning procedure, which is divided into three stages: Picking, Moving, and Placing.

The Picking stage starts with the origin pose obtained by the box acquisition model, where a position and an orientation are defined for the end effector to pick the box, along with a joint configuration that will reach said position and orientation. The Moving stage executes the sequence of movements that make up the path of the robotic manipulator. It starts with calculating the center of the box to move the gripper to the position of the box, hold it, lift it, and move it toward the conveyor belt. It is configured to connect the Picking and Placing positions in an ergonomic way for the manipulator. Finally, the Placing stage consists of lowering and dropping the box on the belt in a fixed position and returning to the identification pose of the box, completing a depalletizing cycle per box. This cycle is repeated until the camera detects the empty pallet, indicating that the process has finished, using a message.

### 3.2. Box Acquisition

Imaging sensors provide little inherent understanding of the scenes they capture. In the box acquisition stage, the boxes must be identified in both the projected UV space defined by pixels and the three-dimensional coordinates. To achieve this, the following distinct problems must be addressed: box detection, pose estimation, pose estimation, and hand–eye calibration.

#### 3.2.1. Box Detection

Locating an object of interest within a digital image by estimating the coordinates and size of its bounding box is a computer vision task known as object detection. Given that boxes on pallets are typically near each other, bounding boxes can only separate different objects in the scene if they are rectangular and have their horizontal and vertical axes aligned with those of the image. This also assumes that no occlusion occurs, as boxes that overlap each other make feature matching more challenging. To address the boxes where one or several of the conditions cannot be guaranteed, the box detection algorithm must be able to identify either non-rectangular polygons or specific pixels corresponding to each object detection. This is formally known as an instance segmentation problem. Examples of successful instance segmentation results are displayed in Figure 2.

We employ the detection and segmentation architectures proposed by Ultralytics on their YOLO series of models [29] trained on synthetic data. A portion of the mentioned data intentionally replicate challenging lighting conditions, with cluttering and glare in particular being factors that strongly influence the performance of the algorithm. A total of 1500 image sets replicating the data captured by the camera array (left grayscale, right grayscale and center color) were generated and then split into train/validation/test following a 60/20/20 ratio. Samples of generated data can be seen in Figure 3. Each detection is used to extract a cropped patch from the image that serves as an input for the next stage of visual processing.

#### 3.2.2. Pose Estimation from Corner Position

We propose a heatmap-based corner detection algorithm inspired by the pallet center detection method presented by Nvidia, where synthetic data are used to train a UNet segmentation model with a Resnet-18 backbone. The model architecture used in our work is presented in Figure 4. The segmentation model was trained for 100 epochs on the expanded crops produced by the detection step.

A low-resolution pass through the segmentation network produces coarse predictions for the corner locations that are then used to formulate region proposals, where the pixel prediction is refined by combining the heatmap output with traditional edge detection procedures. This produces a two-dimensional array where each value represents the likelihood of the indices for said value to correspond to the UV coordinates of a pixel containing one of the four corners of the topmost face. As each corner is detected, the array is masked on its UV coordinates so that nearby pixels of similar intensity are not confused for a new corner. This procedure is repeated for the RGB image captured by the central image sensor as well as the greyscale infrared images from the stereoscopic array. A simple ordering algorithm based on the angle of vectors for each corner relative to the centroid of all corners allows for matching the features on all images. Examples of corner heatmap predictions and refined corner position estimations are shown in Figure 5.

With the variation of the relative *x* pixel coordinates for each corner between the left and right infrared images, it is possible to calculate a sparse disparity value, which can then be related to a numerical depth value according to the relationship presented in Equation (Equation 1).(1)$$depth\left[mm\right]=\frac{f_{x}\left[px\right]*rv\left[mm\right]}{disparity[px]}$$

In this equation, $f_{x}$ is the focal distance on the horizontal plane expressed in pixels, obtained from a camera calibration procedure, and rv is a reference value given by the distance between the two cameras, which for the OAK-D Lite is equal to 75 mm. The greater the disparity, the closer the distance to the camera. With known depths, it is possible to obtain the three-dimensional coordinates for each corner relative to the camera, which can then be used to obtain the absolute size of the topmost face of the box and the coordinates of its center.

#### 3.2.3. Hand–Eye Calibration

To obtain the motion plan in which the robot picks up the box from the pallet, the coordinates and orientation must be expressed relative to the coordinate frame of the base of the robot, not relative to the camera. The problem of finding a homogeneous transformation matrix $H_{cam}^{ee}$ that transforms the coordinates with respect to (w.r.t) the camera to coordinates w.r.t the robotic manipulator is known as the hand–eye calibration problem. This variation, where the camera is mounted on the end effector, is called eye-in-hand calibration. Once obtained, the calibration matrix will be valid as long as the configuration of the camera relative to the end effector remains stationary. Marker-based procedures consist of capturing physical patterns from several different poses selected by a human worker; the discernible key points of the pattern remain visible but the viewpoints are also sufficiently dissimilar to reduce ambiguity when solving for the $H_{cam}^{ee}$ and $H_{object}^{cam}$ in Equation (Equation 2).(2)$$H_{object}^{robot}=H_{cam}^{robot}*H_{object}^{cam}=H_{ee}^{robot}*H_{cam}^{ee}*H_{object}^{cam}$$

We propose a flexible method that simplifies the calibration procedure and does not rely on externally calibrated patterns. As three-dimensional data are already available from the corner position estimation step, the number of captures needed to solve for all variables in Equation (Equation 2) is reduced. This also allows for the automation of the acquisition of new viewpoints, requiring only a static box that fits entirely within the field of view of the robot. After an initial capture of 3D coordinates, it is assumed that the pose of the box w.r.t the end effector is exactly equal to the pose of the box w.r.t the camera. A series of known transformations are applied equally to both the camera and the end effector by moving the robot to different configurations that alter both the position and the orientation of the end effector. Considering the known transformation matrices, the system results in a matrix equation of the form:(3)AX=XB

*A* and *B* are known transformations, from the base of the robot to the end effector and from the detected boxes to the origin of the camera, respectively, while *X* is the unknown transformation from the camera to the end effector: the hand–eye calibration matrix. This is a particular form of Sylvester’s equation, and for transformation, matrices can be solved using methods such as Tsai–Lenz. After obtaining the hand–eye calibration matrix, the transformation of the box with the highest *z* coordinate is set as the origin pose for the path planning stages.

### 3.3. Estimating the End Effector Pose in Depalletizing Processes

In depalletizing scenarios where the boxes that compose the pallet are not aligned with the coordinate axes of the robotic manipulator base, it is crucial to estimate the rotation angle necessary to align the end effector plane of the manipulator with the plane of the box. This alignment ensures the best possible pose to effectively grasp the box and carry out the depalletizing process using pick-and-place techniques. Below, we present the pseudo-code of the proposed Algorithm 1 to estimate the position of the end effector in a depalletizing process from a heterogeneous pallet.
**Algorithm 1** Box orientation estimation     **Input:** $C_{b}$ (4 × 3 box corner coordinate matrix)     **Output:** $R_{b}$ (3 × 3 rotation matrix defining box orientation)1:$P_{r}\leftarrow \left[0,0,1,0\right]$▹ Define reference plane at the origin (z = 0)2:$M_{b}\leftarrow subdivide\left(C_{b}\right)$▹ Generate plane mesh points3:$P_{b}\leftarrow least\_squares\left(M_{b}\right)$▹ Compute plane equation for the box surface4:$V_{r}\leftarrow \left(P_{r}\times P_{b}\right)$▹ Compute rotation vector from cross-product5:$R_{b}\leftarrow \mathrm{Rodrigues}\left(V_{r}\right)$▹ Convert rotation vector to rotation matrix6:**return** $R_{b}$

The first step is to establish the horizontal reference plane $P_{r}$ (line 1). Next, the plane corresponding to the top face of the box is defined using the mesh $M_{b}$, generated from a 3 × 3 grid and the vertices of the box (line 3). Then, the normals of planes $P_{r}$ and $M_{b}$ are determined, and the axis of rotation is calculated using the cross-product between these normals (line 4). Finally, this rotation vector is normalized to calculate the rotation matrix that will allow the planes to be aligned using Rodrigues’s formula (line 5).

To estimate the angle of rotation between the plane of the box and the plane of the base of the robotic manipulator, the first step is to define the planes from the *x*, *y*, and *z* coordinates of the top vertices of the box corners. With these vertices, additional points have been calculated that form a 3 × 3 grid, called “the initial points”. Simultaneously, a reference plane is established parallel to the robot base plane, taking as reference the first upper vertex of the box. From the dimensions of the box, a 3 × 3 grid orthogonal to the axes of the robotic manipulator, called “the endpoints”, is calculated. Each of these two sets of points represents a plane, and the objective is to find the rotation angles on the different axes between these two planes.

The next step in finding these angles is to estimate the constants of each of the planes using a system of linear equations based on the least squares method. This method minimizes the sum of squares of the differences (residuals) between the values predicted by the plane and the actual values, thus fitting the parameters of the plane to the observed points. The general equation of the plane is:(4)Ax+By+Cz+D=0
Next, the normal vectors of the planes are calculated through the cross-product of vectors. This normal vector is essential to determine the required rotation; in case the planes are parallel (i.e., when the normal vectors are collinear), the angle of rotation will be zero.

Once the rotation vector needed to align the planes has been determined, Rodrigues’s formula is used to obtain the rotation matrix around the axis defined by the vector and the calculated angle. See Equation (Equation 5). It is important to take into account the order of multiplications of the rotation matrices, since rotations in three-dimensional space do not commute (the order affects the final result) [30,31].

Rodrigues’s method is used to find the rotation matrix in three dimensions from a rotation vector and a rotation angle. This technique is particularly useful when one wishes to rotate a vector or an object around an arbitrary axis in 3D. In Figure 6, the rotation axis is obtained by the cross-product between the normal vectors of the initial and final planes. If the angle between the two normal vectors is zero, it means that the vectors are collinear [32].(5)$$v^{\prime }=v*cos\left(\theta \right)+\left(k\times v\right)*sen\left(\theta \right)+k*\left(k\times v\right)*\left(1-cos\left(\theta \right)\right)$$
where **v′** is the resultant vector after rotation, *k* is the axis of rotation, a unit vector, θ, is the angle of rotation in radians, and *k* × *v* is the cross-product. Rodrigues’s formula can also be expressed in a matrix formula:(6)$$R\left(\theta \right)=I+sin\left(\theta \right)*K+\left(1-cos\left(\theta \right)\right)*K^{2}$$
where R(θ) is the resultant rotation matrix, *I* is the 3 × 3 identity matrix, and *K* is the matrix of the vector version of the cross-product of the rotation axis k, also known as the antisymmetric matrix or Lie matrix, given by:(7)$$K=\left|\begin{array}{ccc} 0 & -kz & -ky\\ kz & 0 & -kx\\ -ky & kx & 0 \end{array}\right|$$
Here, kx, ky, and kz are the components of the vector *k*.

To validate the obtained rotation matrix, the rotation angles are decomposed using Euler’s convention in X−Y−Z order. Subsequently, a reverse process is applied, where the calculated rotation angles are used to reconstruct the rotation matrix and, consequently, the homogeneous transformation matrix. These calculated angles are sent to the robotic manipulator to orient the end effector so that it is parallel to the box surfaces, thus ensuring efficient and accurate depalletizing.

### 3.4. Motion Planning

The motion planning process for the automated depalletizing system relies on the definition of a series of specific end effector poses that serve as waypoints in a blended trajectory. Successive waypoints are constructed from the location of boxes and known static geometry information of the workspace to avoid collisions. Once the vision module correctly identifies a box and estimates its center, it signals the path planning module to start the planning procedure. This procedure combines the PRM (Probabilistic Roadmap) algorithm from [33] and the A* algorithm as implemented in [34], optimized for the static environment of robotic depalletizing. Initially, the PRM generates a configuration graph by sampling random points in the workspace of the manipulator, discarding those that collide with obstacles such as nearby boxes or the operational area boundaries. Valid points are connected using the CKDTree nearest neighbors search algorithm, forming an undirected graph that represents the robot’s possible trajectories. From this graph, the A* algorithm is employed to find the shortest path between the picking and placing positions, avoiding collisions, and considering cumulative costs.

Once the path is computed, it is transformed from the Cartesian space to the robotic configuration space of the manipulator using inverse kinematics, which results in a sequence of waypoints. These waypoints represent critical intermediate positions through which the manipulator must traverse to complete the planned trajectory. The PRM-generated graph can be reused across multiple cycles without recomputing. Additionally, combining PRM and A* allows adaptation to different pallet configurations and static obstacles. A grid search process is performed to fine-tune the PRM planner parameters (such as the number of samples, connection radius, and the number of neighbors). In each iteration, a set of parameters is generated from a pre-determined distribution and then used to construct the PRM graph. Following this, A* is used to attempt to find a viable path. If a path is found, its cost is calculated based on the sum of the number of samples and the path length. If this cost is lower than previous ones, the parameters and path are saved. The proposed approach enhances sampling, as explained in [35], by directing nodes toward the goal. This involves guided sampling to generate more points near the initial, final, and intermediate points, reducing the execution time of the PRM.

## 4. Results

The results obtained are presented below and are structured in three main stages. First, the robotic cell configuration system used to carry out the experimental test is described. In the second stage, the virtual and experimental tests performed to validate the accuracy of the artificial vision methods used in locating and estimating the position of the boxes are detailed. Finally, in the third stage, the depalletizing path is presented, divided into the stages defined in the “Proposed Depalletizing Scheme” section, together with the analysis of the variables of time, distance, and average linear velocity. In addition, the behavior of the segmented path for each coordinate axis in the depalletizing process of a box is examined.

### 4.1. System Configuration

The system incorporates a UR10 robot, together with a VGC 10 gripper and an OAK-D Lite RGB-D camera attached to the robot end effector link. The configuration of the system is limited to holding the boxes by the topmost face; this represents a reduction in the possible solutions of the grasping problem.

The parallel slit of the complete pallet is 40 × 34 × 130 cm. The pallet is composed of nine levels of boxes, where each different level can exhibit one of the palletization sequences shown in Figure 7. The dimensions and the weight of the boxes are presented in Table 1.

The position units have been defined in meters, and the Cartesian coordinates corresponding to the positions of the boxes and the end effector are expressed relative to the frame of reference of the base of the robotic manipulator. This choice aims to standardize the process between the various modules that make up the system. A view of the system may be found in Figure 1a. The origin of the frame of reference (x,y,z,rx,ry,rz) is located at the base of the robot.

### 4.2. Box Acquisition

To validate the accuracy of the computer vision methods used to locate and estimate the pose of boxes from data acquired by the RGB-D sensors, both virtual and experimental tests were performed. For the object detection task, box recall and accuracy were measured for a validation dataset constructed in the same randomized manner as the training dataset, where the detection model achieves perfect mean average precision scores. No false positives or false negatives were encountered in real-world detection experiments.

#### 4.2.1. Corner Detection

In the box of the corner position estimation step, the proposed heatmap regression network was pitted against FAST-CPDA [36], a corner detector based on morphological operations. The latter was chosen according to the performance evaluation presented by Zhang et al. [37], as it offered a reasonable balance between speed and accuracy, even when compared to more recent corner detection methods.

Our heatmap regressor and the FAST-CDPA algorithms were used to detect corner positions on 1000 images that had already been cropped to the region of interest containing boxes with an additional border margin of 20 pixels. All cropped images were scaled to the regressor input size of 512 × 512 pixels. Unlike the regressor, which by design will always output four corner coordinates (resulting in precision and recall being equal), FAST-CDPA occasionally produces a greater or lower number of corner detections. Whenever the distance between any ground truth corner and the nearest detected corner was greater than 5 pixels (the size of the Gaussian kernel used to produce ground truth heatmaps), a false negative was recorded. In a similar manner, every detected corner that is more than 5 pixels away from all ground truth corners was registered as a false positive. Average position error is computed only for true positives, and is reported along with precision and recall for each method in Table 2 for both methods.

Under the mentioned conditions, the average position error for the UNet-based heatmap regressor was 1.26 pixels, while FAST-CDPA exhibited an average position error of 0.91 pixels. However, precision and recall performance are significantly lower for the FAST-CDPA algorithm. Average computation time was 52 ms for the heatmap regressor while running on an NVidia Quadro T600 GPU; FAST-CDPA on average took 36 ms on an Intel Core i7-12700 CPU. These results show that the proposed method achieves competitive corner detection performance with FAST-CDPA. From the results presented in Figure 8, it can be inferred that the proposed method is more robust to lighting disturbances, irregular box edges and clutter.

#### 4.2.2. Box Pose Estimation

The accuracy and precision of three-dimensional corner position estimation were evaluated by placing a box in a fixed position and manually configuring the robot so that the gripper was aligned with the center of the box. As it was the box most prone to orientation errors due to having the shortest edges, box c1 was chosen for this experiment to evaluate performance in a worst-case scenario. Joint angles were recorded on this configuration to obtain a ground truth for the transformation of the box relative to the rest of the robot.

Following the acquisition of the ground truth orientation and position, these were estimated by placing the robot in 12 different configurations that ensured significantly different orientations and translations of the end effector and sensor relative to the box. These configurations resulted in multiple recorded viewpoints, shown for the left infrared sensor in Figure 9, which approximates the views of boxes that could feasibly be encountered in a heterogeneous pallet.

From these viewpoints, the size and pose of the box were estimated using the segmentation network, the disparity-to-depth algorithm, and Rodrigues’s formula. Table 3 summarizes the results of this experiment.

The average position error is 2.45 cm, while the average orientation error is 2.76 degrees. These results communicate the accuracy of both the box pose estimation algorithm and the hand–eye calibration algorithm. The former is in line with passive stereo techniques used in depalletizing systems and is within the acceptable tolerance of vacuum grippers like the VGC10 [24], while the latter is comparable to other automated hand–eye calibration procedures using closely spaced stereo cameras [38]. The processing of the images through the detection and segmentation model takes 62 ms on average on a computer running a Linux-based operating system with a Core i7-12700 CPU, 64 GB of DDR4 RAM an NVidia Quadro T600 with 4 GB of GDDR6 VRAM. This equates to approximately 15 frames per second (FPS), considerably faster than recent approaches based on dense depth [15] that operate between 1 and 2 FPS.

### 4.3. Motion Planning

The experimental test in the laboratory was carried out taking into account the motion stages established in the Proposed Depalletizing Scheme, which are picking, moving, and placing, as illustrated in Figure 10.

#### 4.3.1. Velocity and Acceleration Limits

To define the kinematic limits for acceleration and velocity of the robotic manipulator, a sensitivity analysis was conducted. In this exercise, the distance variable was fixed, while the angular velocity limit for every joint was set to values between 0.5 and 3 rad/s and the angular acceleration to values between 0.1 and 2.1 rad/s². As shown in Figure 11, when varying acceleration within the established range, the time curve exhibited an analogous behavior regardless of the changes in velocity and acceleration. From an acceleration of 1.5 rad/s² onward, the trajectory execution time stabilizes. Regarding velocity, no significant differences in time behavior were observed for values of 2, 2.5 and 3 rad/s. However, it was found that velocities above 2.5 rad/s resulted in abrupt acceleration and deceleration of the joints. Consequently, an angular velocity of 2 rad/s was established, achieving the required operational stability.

#### 4.3.2. Waypoint-Based Path Planning

A simulated experiment in a virtual environment was performed to establish a comparison between the basic PRM path planning algorithm combined with A* and an improved version using directed sampling. The initial parameters were explored using the following limits; the number of samples ranged from 100 to 200, the connection radius from 0.05 to 2, and the number of neighbors from 3 to 50. Fifty replicates were performed for each algorithm. The results are presented in Table 4.

For the basic PRM A*, three waypoints were generated to define the trajectory, with an average time of 11.59 s. Meanwhile, for the improved PRM A*, five waypoints were created in an average time of 8.49 s. The PRM used parameters generated by the improved algorithm, with the number of samples set to 100, a connection radius of 0.96, and 26 neighbors. Path planning used these values in subsequent experiments.

#### 4.3.3. Traveled Distance and Path Execution Time

The trajectory is established in Cartesian space concerning the coordinate system located at the base of the robotic manipulator. During the depalletizing exercise, it was observed that the displacements of the end effector are distributed on the x-axis from −0.11 to 0.07 m, on the y-axis from −0.88 to 0.62 m, and on the z-axis from −0.7 to 0.6 m. The mean traversed distance was 2.82 meters, with trajectories completed in 6.73 s on average, excluding the time spent activating and deactivating the gripper. More time and distance data are available in Appendix A Table A1.

Figure 12 illustrates the behavior of the segmented path on each coordinate axis. In the graphs of the positions on the *x* and *z* axes, it is possible to identify the picking and placement stages on the plateaus, representing the moments in which the end effector remained stationary. It can also be observed that the position on the *y* axis remains constant, suggesting a reduction in the distance traveled.

## 5. Conclusions

Automated depalletizing of heterogeneous cargo requires a sufficient understanding of the workspace and path planning adapted to box surfaces exhibiting arbitrary orientations concerning the coordinate frame of the robot. We introduce a machine vision algorithm that can accurately retrieve three-dimensional pose and semantic information from sparse stereo-depth information obtained from RGB-D images. This algorithm detects and identifies box dimensions, position, and orientation at a faster rate than methods that rely on processing dense depth data, with greater adaptability to challenging lighting conditions and occlusions than high-performing morphological methods. We also propose a path-planning trajectory method based on probabilistic roadmaps that can perform automated depalletizing when boxes are stacked at tilted angles, implementing a grasp design procedure that accounts for their orientation and obstacles resulting from pallet geometry. The algorithms presented in this paper were validated experimentally on a robotic work cell composed of a UR10 robotic manipulator, a vacuum end effector (VGC10 gripper), an OAK-D Lite camera, and a conveyor belt. The results showed that the vision algorithm produces an average position error of 2.45 cm at distances ranging from 40 cm to 150 cm, displaying enough accuracy to successfully deconstruct pallets using the proposed path planning strategy. This strategy uses Rodrigues’s formula to adjust the gripper orientation according to the orientation of the topmost face of a box while using known robot constraints to reduce the computational complexity of path planning calculations.

## Figures and Tables

**Figure 1 sensors-25-01137-f001:**
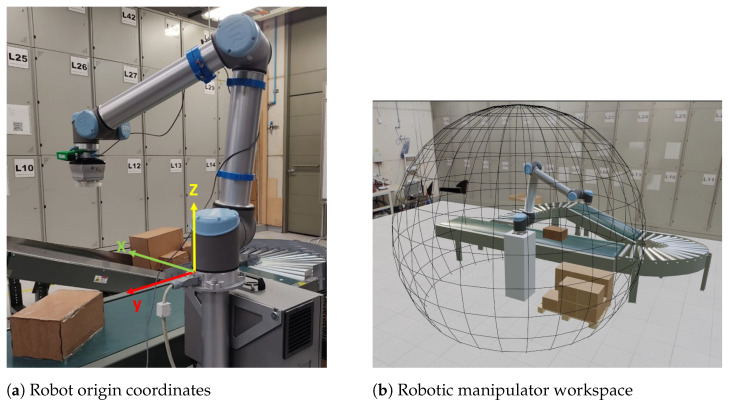
Workspace definition.

**Figure 2 sensors-25-01137-f002:**
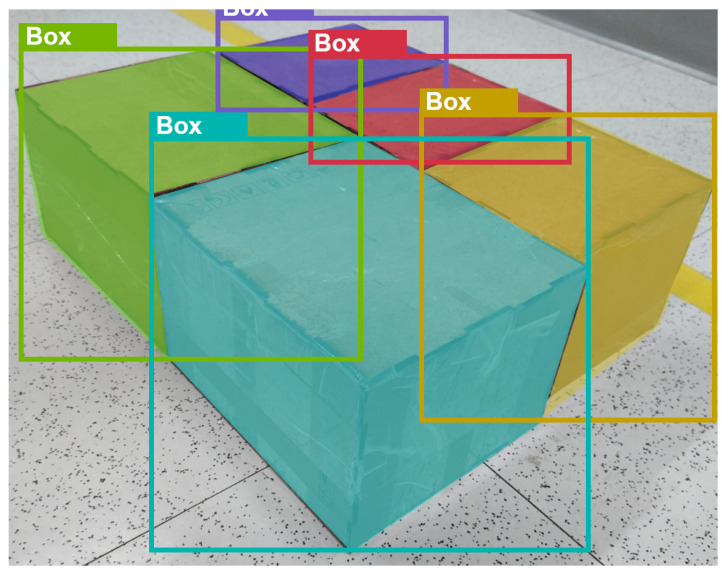
Segmentation results for a non-orthogonal view of a constructed pallet.

**Figure 3 sensors-25-01137-f003:**
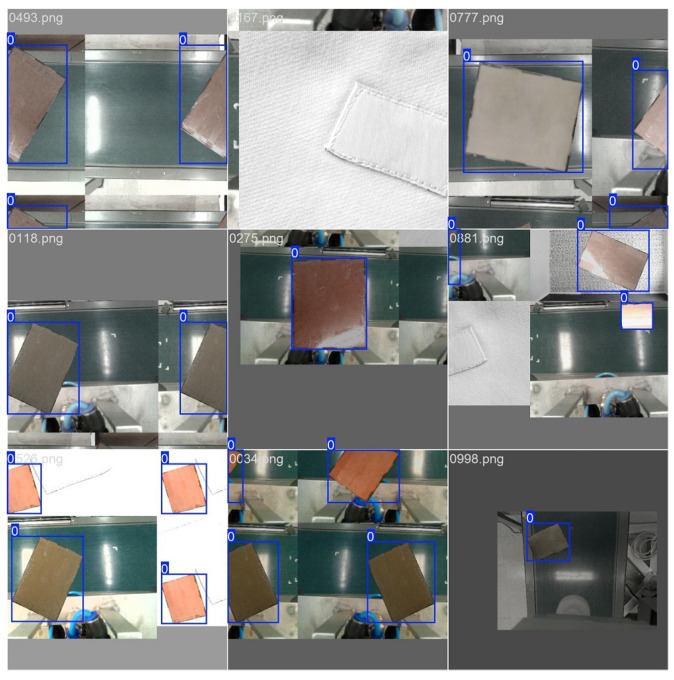
Synthetic data generated for the purpose of training the object detection and segmentation algorithm. Detection bounding boxes are drawn in blue.

**Figure 4 sensors-25-01137-f004:**
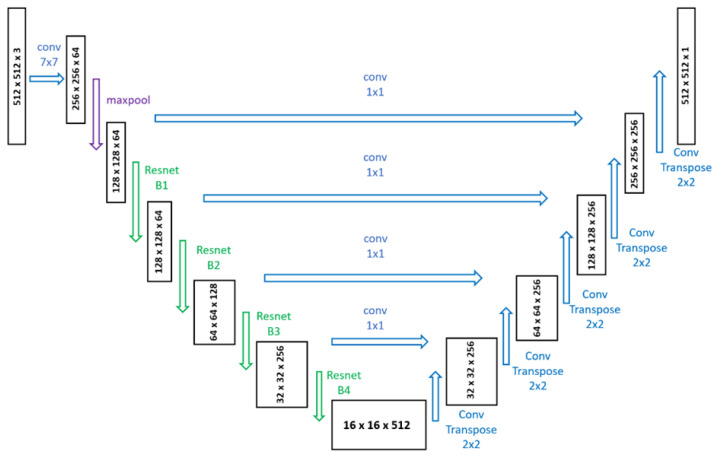
UNet architecture with a ResNet-18 backbone used in this paper to perform heatmap-based corner predictions.

**Figure 5 sensors-25-01137-f005:**
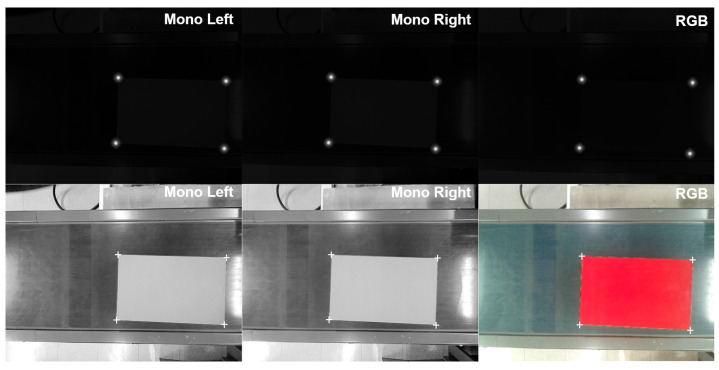
Heat map predictions for corner locations (**top**) from input images captured by the camera array (**bottom**). Brighter pixel values indicate a higher probability of a corner existing at the related coordinates.

**Figure 6 sensors-25-01137-f006:**
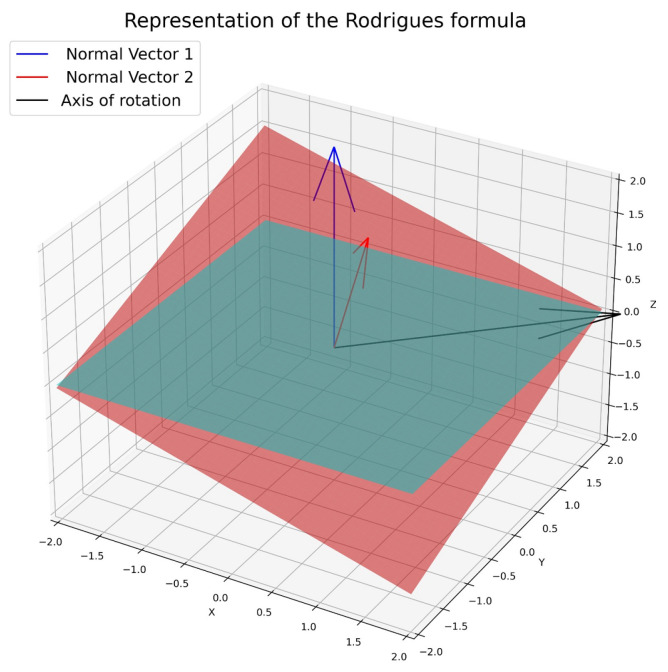
Visual representation of Rodrigues’s formula. The cross-product of the normal vector for the red plane by that of the horizontal plane results in the axis of rotation for the plane, displayed as the black vector on the figure.

**Figure 7 sensors-25-01137-f007:**
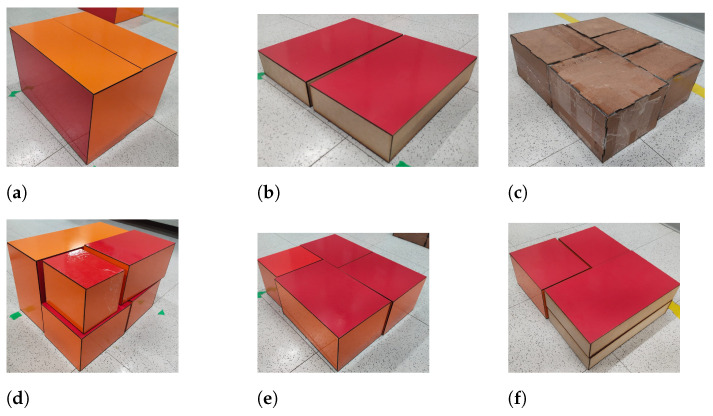
Different types of levels expected during depalletizing. Starting clockwise from (**a**–**f**) every level increases in heterogeneity.

**Figure 8 sensors-25-01137-f008:**
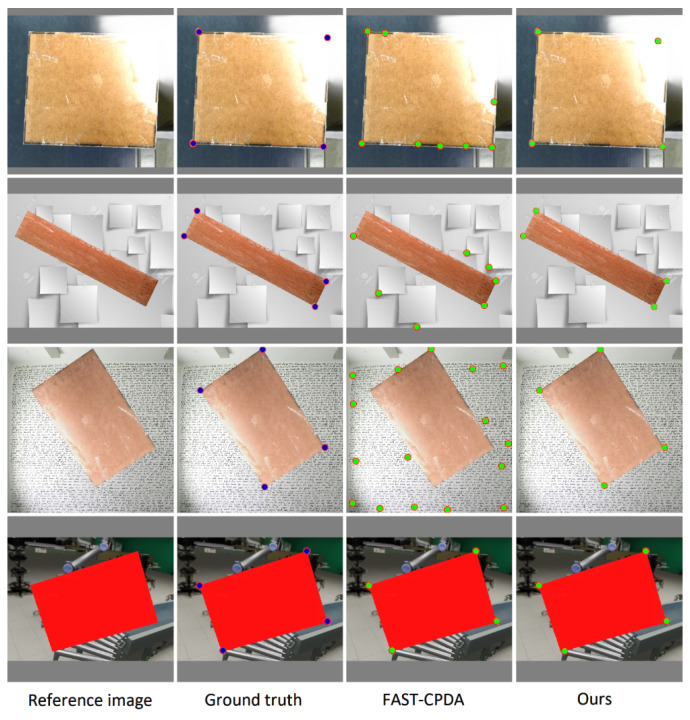
Qualitative comparison of our corner detection method with the high-performing FAST-CDPA. Blue circles represent the ground truth and green circles represent the predictions.

**Figure 9 sensors-25-01137-f009:**
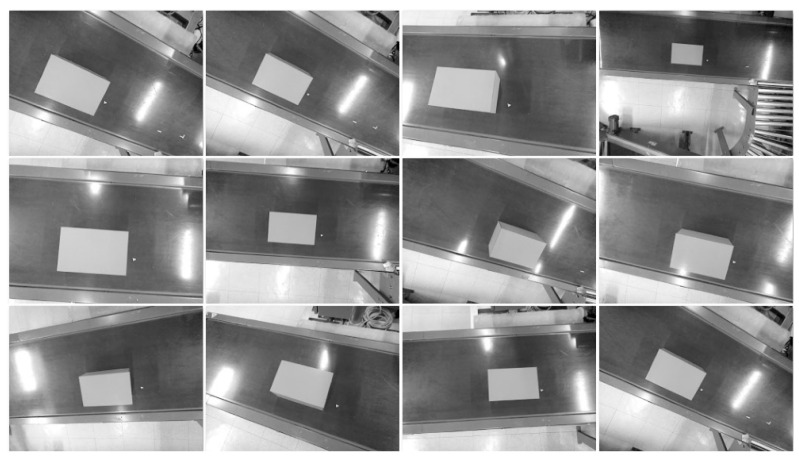
Different viewpoints used in the pose estimation experiment.

**Figure 10 sensors-25-01137-f010:**
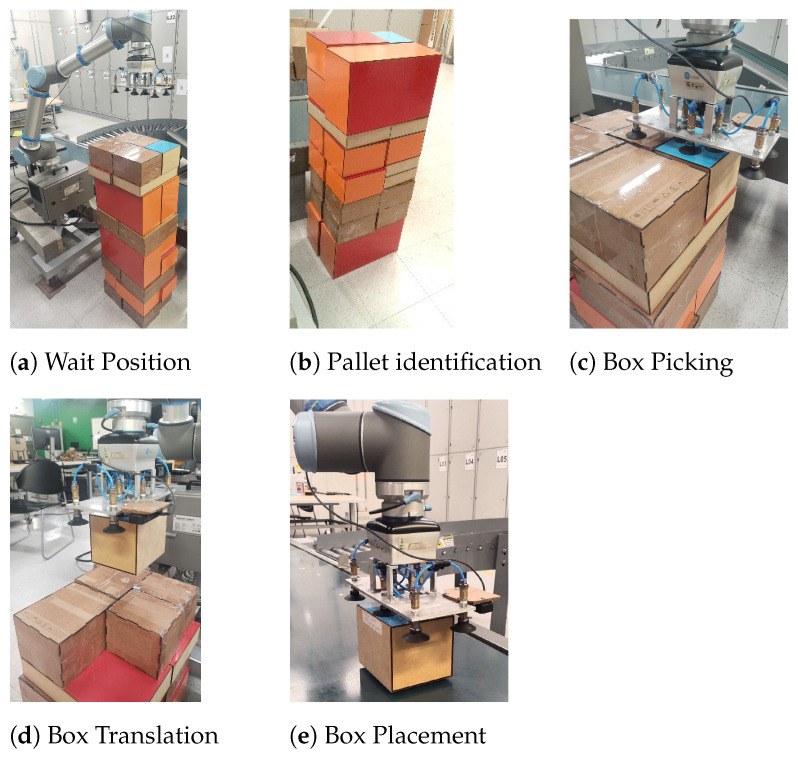
Stages of the trajectory.

**Figure 11 sensors-25-01137-f011:**
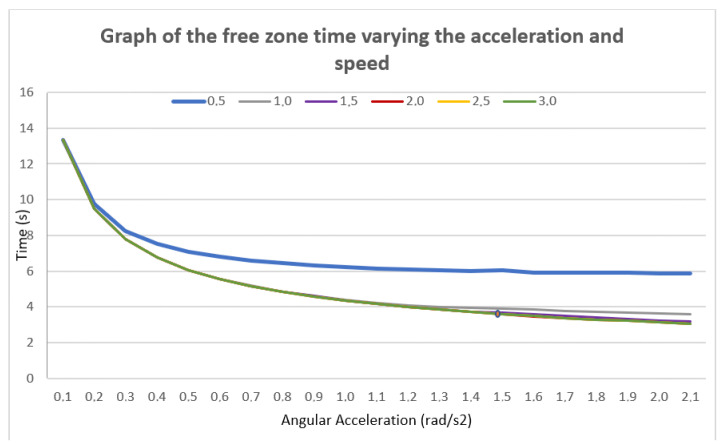
Speed and acceleration sensitivity test.

**Figure 12 sensors-25-01137-f012:**
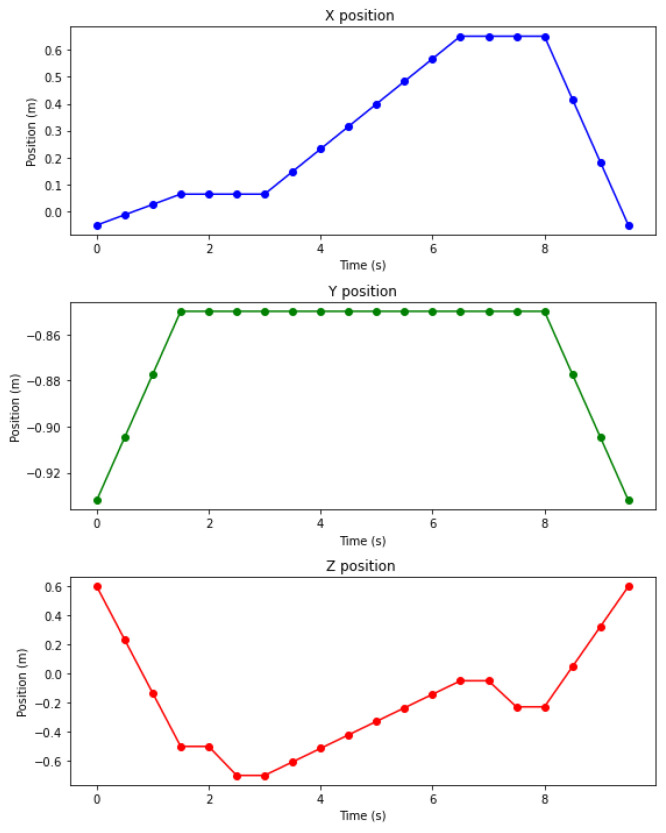
Evolution of end effector position by axis while reaching different waypoints.

**Table 1 sensors-25-01137-t001:** Box specifications (all in centimeters).

Box	Width	Length	Height	Quantity	Weight [g]
c1	12	15	12	10	326.2
c2	16	19	12	9	474.6
c3	15	39	24	3	1262.8
c4	16	25	12	10	588.0
c5	19	32	6	4	600.6

**Table 2 sensors-25-01137-t002:** Corner detection metrics.

	FAST-CPDA	Ours
Average position error [px]	**0.91**	1.26
Precision	0.791	**0.939**
Recall	0.837	**0.939**
Average processing time [ms]	**36**	52

**Table 3 sensors-25-01137-t003:** Position and orientation error.

Pose	Position Error [m]	Orientation Error [°]
1	0.0183	3.026
2	0.0091	2.6753
3	0.0223	4.8284
4	0.0446	3.3983
5	0.1091	2.1888
6	0.0207	2.2291
7	0.012	1.4445
8	0.0081	2.7595
9	0.0094	0.8262
10	0.0045	5.352
11	0.0128	2.1061
12	0.0238	2.3623
Average	0.0245	2.7663

**Table 4 sensors-25-01137-t004:** Comparison of PRM A* Basic and Improved for the estimation of PRM parameters.

Distribution	Num Samples	Connection Radius	Num Neighbors	Time (s)	Total Samples Generated	Waypoints
Random centered Basic	100	1.14	26	11,59	120	3
Improved Random	100	0.96	26	8.49	135	5

## Data Availability

The original data presented in the study are openly available at https://github.com/jcmartinez10/SEMA (accessed on 6 December 2024).

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
