# Peer review of "Machine Vision-Assisted Design of End Effector Pose in Robotic Mixed Depalletizing of Heterogeneous Cargo"

_sensors, 2025, doi:10.3390/s25041137_

Round 1
Reviewer 1 Report
Comments and Suggestions for Authors
1. Problems of the current situation: It is suggested that the core key problems of the system should be summarized in the introduction, and it can be seen from the method of this paper that they are divided into visual positioning problems, calculation inclination problems, and path planning problems, and the existing literature should describe these problems in the current situation.
2. Keywords should focus on the method of the paper itself, do not use too broad words, and try not to use Automation; Computer vision; This kind of big term, which will reduce the search efficiency of keywords.
3. The formula and drawings are not clearly expressed: for example, the naming of Figure 5 is recommended to explain the meaning of the picture. The problem with formulas, such as equation (3), does not explain any variables.
4. From the paper, this paper proposes a new visual localization strategy, which uses the method of object detection and segmentation for recognition, and the corner prediction method of heat map for Pose estimation, which does not clearly express the principle and advantages of the method in this paper. In addition, how the advantages of these two methods are proved is not shown in the experimental part. It is suggested that the principles and advantages of this method should be expressed in detail, and it is best to prove them through comparative experiments.
5. In the same way: the algorithm of the path planning part should also be fully expressed, and comparative experiments should be added.
Comments on the Quality of English Language
There is still some room for improvement in English expression skills. It is recommended that authors pay more attention to the accuracy and fluency of language when writing papers, and avoid using words and sentences that are too complex or obscure, so as to improve the readability and understandability of the paper.
Author Response
- Comment 1: Problems of the current situation: It is suggested that the core key problems of the system should be summarized in the introduction, and it can be seen from the method of this paper that they are divided into visual positioning problems, calculation inclination problems, and path planning problems, and the existing literature should describe these problems in the current situation.
- - Response 1: We agree with the suggested revisions, and we have modified the introduction section to include the necessary context. Additions can be found on lines 27-38.
- - Comment 2: Keywords should focus on the method of the paper itself, do not use too broad words, and try not to use Automation; Computer vision; This kind of big term, which will reduce the search efficiency of keywords.
- - Response 2: We have proposed new keywords that we believe better represent the area of interest of this paper.
- - Comment 3: The formula and drawings are not clearly expressed: for example, the naming of Figure 5 is recommended to explain the meaning of the picture. The problem with formulas, such as equation (3), does not explain any variables.
- - Response 3: Additional context has been added to figures 3, 4, 5 and 6: An explanation for variables present in equation 3 has been added on line 244.
- - Comment 4: From the paper, this paper proposes a new visual localization strategy, which uses the method of object detection and segmentation for recognition, and the corner prediction method of heat map for Pose estimation, which does not clearly express the principle and advantages of the method in this paper. In addition, how the advantages of these two methods are proved is not shown in the experimental part. It is suggested that the principles and advantages of this method should be expressed in detail, and it is best to prove them through comparative experiments.
- - Response 4: We agree with the point raised by the reviewer. Comparative experiments were performed in comparison with other visual processing strategies such as FAST-CDPA. We show how our method is resilient to false positive predictions in cluttered scenes and to false negative errors in challenging lighting conditions, presenting both qualitative and quantitative comparisons with the aforementioned method. These additions can be found on section 4.2.1. -
- Comment 5: In the same way: the algorithm of the path planning part should also be fully expressed, and comparative experiments should be added.
- - Response 5: We have expanded upon the algorithm used for the definition of waypoints, and experimental comparisons have been performed. Section 3.4 was rewritten, and comparative results were added to section 4.3.
Reviewer 2 Report
Comments and Suggestions for Authors
I am honored to review this manuscript. The study explored real-time damage detection networks in mines based on conveyor belts with knowledge distillation. However, I believe that the manuscript requires substantial revisions before it can be considered for publication. Below, I outline the key areas that would benefit from further improvements.
1. Units are not indicated in Table 1.
2. Specific details of the datasets, such as training, validation, and test sets, are missing.
3. Performance results of the algorithm (such as precision and recall) are not provided.
4. There is no comparability with other algorithms.
5. The network model of the algorithm is not shown.
6. The weight of the cargo is not mentioned.
7. Consideration of speed as a factor is not addressed in the paper.
8. The names of the images are obscured in the figures.
I recommend that the manuscript undergoes significant revisions to improve its accuracy and overall quality. I encourage the authors to carefully address the points raised above.
Author Response
Comment 1: Units are not indicated in Table 1.
Response 1: Units have been included.
Comment 2: Specific details of the datasets, such as training, validation, and test sets, are missing.
Response 2: More details were included on lines 181-186.
Comment 3: Performance results of the algorithm (such as precision and recall) are not provided.
Response 3: The algorithm presents two different steps in which precision and recall performance metrics can be considered, the bounding box detection step and the heatmap regression step. However, on the first task, the model achieves perfect mAP scores for all training, validation, and test images. This had already been reported on lines 367-370. Results of comparative experiments for the heatmap regression step were included on section 4.2.
Comment 4: There is no comparability with other algorithms.
Response 4: Additional comparisons with techniques based on morphological operations (FAST-CDPA) were performed. Average pixel coordinate errors are reported for reviewed methods. Qualitative results were added on Figure 8.
Comment 5: The network model of the algorithm is not shown.
Response 5: We do not modify the standard YOLOv8 architecture and as such, we refer to its description online for additional model details. Figure 4 shows the model we used to perform corner heatmap regression. Convolution and transpose convolution operations are included, as well as the residual blocks which are implemented as specified on the reference introducing ResNET.
Comment 6: The weight of the cargo is not mentioned.
Response 6: A column detailing cargo weight for each specific type of box was added to Table 1. -
Comment 7: Consideration of speed as a factor is not addressed in the paper.
Response 7: The reasoning behind having a fixed top speed and its value is present on the paper, on lines 432-442.
Comment 8: The names of the images are obscured in the figures.
Response 8: This has been corrected for Figures 6-10.
Reviewer 3 Report
Comments and Suggestions for Authors
1. The manuscript contains several basic errors, such as spelling mistakes, repeated citations of the same source, suspected errors in formulas, and inconsistencies in table and figure formatting. Additionally, the logical flow of the paper has issues, which hinders the clear delivery of the authors' intended message.
2. The methodology section does not sufficiently explain the necessity and superiority of the approaches used in the study. For instance, in Section 3.2.1, the use of YOLO appears to be limited to retraining the data in this specific scenario, lacking innovation and with no subsequent presence in the experimental results. In Section 3.2.3, the mentioned hand-eye calibration method highlights its advantages, but there is no discussion regarding its accuracy. Similarly, the inverse kinematics described in Section 3.4.2 seems to simply rely on the built-in functions of the UR robot without demonstrating the paper's unique contributions.
3. The experimental results appear somewhat insufficient to validate the superiority of the proposed method. I recommend including additional comparative experiments to strengthen the findings.
4. There are many unacceptable written errors, such as "the" in Line 8, "succesfully" in Line 38, "witth" in Line 86 ......
It contains some colloquial expressions and spelling errors. Furthermore, the logical inconsistencies in certain parts make the paper difficult to follow.
Author Response
Comment 1: The manuscript contains several basic errors, such as spelling mistakes, repeated citations of the same source, suspected errors in formulas, and inconsistencies in table and figure formatting. Additionally, the logical flow of the paper has issues, which hinders the clear delivery of the authors' intended message.
Response 1: We have revised the manuscript to correct the errors mentioned by the reviewer, but, if possible, we would like to inquire about the suggested revision regarding repeated citations of the same source. Whenever this occurs in our manuscript, we believe we are referencing different definitions or contributions present on the same paper, which should not contradict MDPI author guidelines.
Comment 2: The methodology section does not sufficiently explain the necessity and superiority of the approaches used in the study. For instance, in Section 3.2.1, the use of YOLO appears to be limited to retraining the data in this specific scenario, lacking innovation and with no subsequent presence in the experimental results. In Section 3.2.3, the mentioned hand-eye calibration method highlights its advantages, but there is no discussion regarding its accuracy. Similarly, the inverse kinematics described in Section 3.4.2 seems to simply rely on the built-in functions of the UR robot without demonstrating the paper's unique contributions.
Response 2: Regarding Section 3.2.1, we do not intend to fill a knowledge gap regarding bounding box-based object detection pipelines like YOLO, but its mention within the paper is necessary given that the proposal of regions of interest is critical to the performance of the corner detection algorithm.
As for section 3.2.3, the experimental results presented on section 4.2 communicate the accuracy of both the pose estimation method and the hand-eye calibration procedure, as the latter is directly derived from the former. Further discussion has been added on lines 417-421.
We agree that Section 3.4 focuses on the technical aspects specific to the control of Universal Robots manipulators at the expense of the theoretical framework of our proposed path-planning algorithm. The section was re-written to account for this insufficiency.
Comment 3: The experimental results appear somewhat insufficient to validate the superiority of the proposed method. I recommend including additional comparative experiments to strengthen the findings.
Response 3: Sections 4.2 and 4.3 have been expanded to include comparative and parameter tuning experiments that better reflect the advantages of the proposed methods.
Comment 4: There are many unacceptable written errors, such as "the" in Line 8, "succesfully" in Line 38, "witth" in Line 86 ......
Response: These and other errors in the manuscript have been corrected.
Round 2
Reviewer 1 Report
Comments and Suggestions for Authors
The paper has been extensively revised, and the overall opinions are as follows:This paper focuses on the research of robotic mixed depalletizing of heterogeneous cargo. The theme is clear and has practical application value. In terms of methods, it combines deep learning with multiple algorithms, innovatively realizing the depalletizing operation in complex scenarios. The experiments are well-verified, and the results effectively prove the feasibility and accuracy of the methods. I agree to the publication of this paper.
Author Response
We appreciate the constructive and insightful comments of the reviewer, that were invaluable in improving the quality of our work.
Reviewer 2 Report
Comments and Suggestions for Authors
The article demonstrates innovative contributions, and the research content possesses considerable practical value, making it well-suited for publication in this journal.
Author Response

(The authors gave the same response as above.)
